# Efficient Federated Prompt Tuning for Black-Box Large Pre-trained Models

## Abstract

With the blowout development of pre-trained models (PTMs), the efficient tuning of these models for diverse downstream applications has emerged as a pivotal research concern. Although recent investigations into prompt tuning have provided promising avenues, three salient challenges persist: *(1) memory constraint*: the continuous growth in the size of open-source PTMs renders fine-tuning, even a fraction of their parameters, challenging for many practitioners. *(2) model privacy*: existing PTMs often function as public API services, with their parameters inaccessible for effective or tailored fine-tuning. *(3) data privacy*: the fine-tuning of PTMs necessitates high-quality datasets, which are typically localized and not shared to public. To optimally harness each local dataset while navigating memory constraints and preserving privacy, we propose **Fed**erated **B**lack-**B**ox **P**rompt **T**uning (**Fed-BBPT**). This innovative approach eschews reliance on parameter architectures and private dataset access, instead capitalizing on a central server that aids local users in collaboratively training a *prompt generator* through regular aggregation. Local users leverage API-driven learning via a zero-order optimizer, obviating the need for PTM deployment. Relative to extensive fine-tuning, Fed-BBPT proficiently sidesteps memory challenges tied to PTM storage and fine-tuning on local machines, tapping into comprehensive, high-quality, yet private training datasets. A thorough evaluation across 40 datasets spanning CV and NLP tasks underscores the robustness of our proposed model.

## 1 Introduction

Large pre-trained models (Radford et al., 2021; OpenAI, 2023a;b; Touvron et al., 2023) have achieved remarkable success across a wide range of downstream tasks. The voluminous parameters enable pre-trained models to capture a broad spectrum of knowledge from massive unlabeled samples, resulting in enhanced fine-tuning performance over models trained from scratch. Nevertheless, the computational overhead remains a formidable barrier, necessitating alternate, efficient solutions. Thus, parameter-efficient transfer learning (PETL) has been developed to tune PTMs, striving for competitive performance via partial parameter fine-tuning. Notably, prompt tuning methods (Gu et al., 2021; Yao et al., 2021; Jia et al., 2022; Han et al., 2022) have emerged as particularly efficacious. These strategies enhance performance in zero-shot or few-shot domains, fully capitalizing on the PTMs' capabilities. Harnessing large-scale models with a limited parameter set has garnered significant attention in artificial intelligence research. Yet, even with these advancements, we confront three principal hurdles in real-world applications:

1. ***Memory Constraint***: contemporary open-source models like GPT-3 (Brown et al., 2020) and LLaMA (Touvron et al., 2023) boast tens or even hundreds of billions of parameters, making it burdensome for users to fine-tune or store them.

2. ***Model Privacy***: the prohibitive costs of training PTMs mean many remain proprietary. Public can only access those PTMs through APIs, such as ChatGPT (OpenAI, 2023b). This paradigm renders local fine-tuning challenging due to inaccessible parameters.

3. ***Data Privacy***: premium datasets are predominantly private. For instance, medical imagery is often exclusive to certain institutions, amplifying challenges tied to data acquisition.

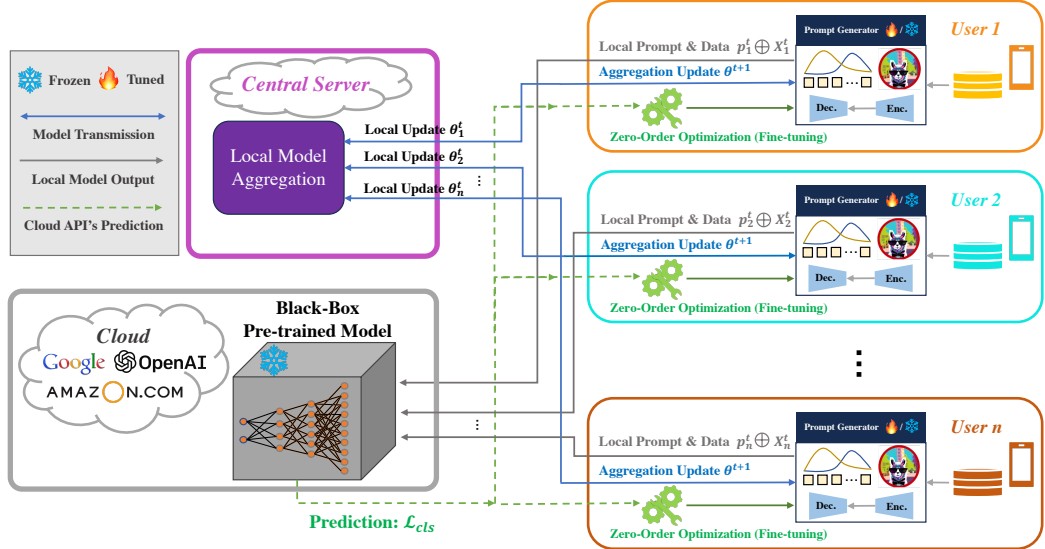

Figure 1: An overview of the proposed framework. Local users adopt a prompt generator linked to the PTM and train it with a black-box optimization. The central server regularly aggregates the local models of all users.

Owing to these constraints, coupled with the bidirectional nature of privacy concerns, PTM applications face significant adoption barriers.

**Motivation.** Spurred by achievements in federated learning (FL) (McMahan et al., 2017) and black-box prompt tuning (BBPT) for large models (Oh et al., 2023; Diao et al., 2022; Chen et al., 2023), we propose the **Fed-BBPT** framework. This design facilitates collaborative, efficient zero-order tuning for PTMs without the constraints of storing large PTMs, accessing model parameters, or sharing proprietary datasets. In juxtaposition with previous federated models like FedGPT (Zhang et al., 2023a) and FedCLIP (Lu et al., 2023), the distinct advantage of Fed-BBPT lies in its independence from local PTM deployment. This significantly saves the storage and makes it conducive for black-box API services. Furthermore, users are liberated from both high computational resources and potential privacy breaches.

We present the proposed paradigm in Figure 1. Local users employ a lightweight prompt generator tailored to their local dataset and connects to the PTMs via API services. Initially, the local user generates appropriate prompts using the current generator and forwards the combined data samples to the PTM's API. Upon receiving the input, the PTM responds with the corresponding output. Due to the absence of first-order gradients in the returned data, we adopt a zeroth-order optimization technique on the local user to evaluate the contribution of the generated prompts for the local task. We employ different strategy depending on Computer Vision (CV) and Natural Language Processing (NLP) tasks, as delineated in Table 2. For each task, after one epoch (or several iterations) of local training, the local model is delivered to the central server for aggregation. Local users can then retrieve the aggregated generator as the initial state for the next training communication round.

We conduct comprehensive experiments on the three CV and NLP tasks, including 12 image classification datasets, 7 text classification datasets from GLUE benchmark (Wang et al., 2018a) and 21 finding best instruction datasets from BIG-Benchmark (Zhou et al., 2022b; Chen et al., 2023). All the experiments yield results that are on par, or surpass, established baselines, attesting to the efficacy of our proposed Fed-BBPT. In summary, the main contributions of this work are:

- We introduce a pioneering federated black-blox prompt tuning framework, Fed-BBPT, which facilitates the joint training of a global lightweight prompt generator across multiple local users with constrained computational resources.

- As delineated in Table 1, our Fed-BBPT framework is the inaugural approach addressing the challenges of memory constraints, model privacy, and data privacy simultaneously.

- Our comprehensive experiments span both the CV and NLP domains and include tasks such as image classification, text classification, and finding the best instruction. The experimental results on 40 datasets affirm the robustness of our proposed framework.

- The consistent results across diverse experimental configurations validate that the Fed-BBTP framework is versatile and can be seamlessly integrated into any black-box tuning scenario and various zeroth-order optimization strategies. This augments the utility of PTMs for local users.

Table 1: Comparison between our proposed framework on all previous baselines.

| Method | Computing Efficiency | Memory Efficiency | Model Privacy | Data Privacy |
|---|:---:|:---:|:---:|:---:|
| Finetune | ✗ | ✗ | ✗ | ✗ |
| FedAvg (McMahan et al., 2017) | ✗ | ✗ | ✗ | ✓ |
| FedCLIP (Lu et al., 2023) | ✓ | ✗ | ✗ | ✓ |
| FedGPT (Zhang et al., 2023a) | ✓ | ✗ | ✗ | ✓ |
| BlackVIP (Oh et al., 2023) | ✓ | ✓ | ✓ | ✗ |
| BDPL (Diao et al., 2022) | ✓ | ✓ | ✓ | ✗ |
| InstructZero (Chen et al., 2023) | ✓ | ✓ | ✓ | ✗ |
| Fed-BBPT (Ours) | ✓ | ✓ | ✓ | ✓ |

## 2  RELATED WORK

**Federated Learning** McMahan et al. (2017) propose the federated learning (FL) framework to jointly train the model across several private heterogeneous datasets. Due to the negative impacts of local training, current studies in FL focus more on the small models, i.e. the ResNet, and Transformer (Li et al., 2020a;b; Zhang et al., 2021). The main challenge focuses on alleviating the inconsistency (Charles & Konečnỳ, 2021; Malinovskiy et al., 2020; Wang et al., 2020) and adopts the proxy function on the local users (Acar et al., 2021; Sun et al., 2023c). Recent advances gradually explore its performance and validate its effectiveness in finetuning large models. Stremmel & Singh (2021) study the pretraining in the federated text models and confirm that joint datasets can further improve performance. Weller et al. (2022); Cho et al. (2022) learn the ensemble knowledge could help the large models to be more stable. Chen et al. (2022) expand its utilization to the CV community. Similarly, Su et al. (2022); Guo et al. (2023); Li et al. (2023a) learn that finetuning visual prompts could efficiently improve the final test accuracy. FL shows excellent performance in the field of optimization with a small dataset, which is exactly what the large models require for efficient finetuning on the limited high-quality dataset. However, the most current FL approaches on large PTMs require white-box PTMs deployed in local users, such as LLaMA (Touvron et al., 2023), but in many situations, large PTMs is close-source serving as black-box APIs, i.e. ChatGPT (OpenAI, 2023b), GPT-4 (OpenAI, 2023a), and Claude-2 (Anthropic, 2023).

**Prompt Tuning** Due to the large cost of the time-consuming finetuning on large PTMs, parameter-efficient tuning methods such as prompt tuning (Lester et al., 2021) and prefix tuning (Li & Liang, 2021) are wildly applied to adapt the large PTMs to target small downstream tasks. In NLP tasks, prompt tuning can be divided into two classes: discrete prompts (Gao et al., 2020; Shin et al., 2020; Ben-David et al., 2022) which is a sequence of discrete tokens and continuous prompts (Zhong et al., 2021; Han et al., 2022; Qin & Eisner, 2021) which is a sequence of continuous vectors followed by input text vectors. Liu et al. (2023) summarize the main developments on prompt learning and give a positive answer to confirm its effectiveness. While in CV tasks, usually image classification or vision-language generation, the prompt is always continuous format (Zhou et al., 2022a; Jia et al., 2022; Sun et al., 2023a). However, the same issue that those methods are limited to white-box accesses happens again. To tackle such problem, the black-box prompt tuning is proposed (Oh et al., 2023; Diao et al., 2022; Chen et al., 2023), which utilizes zeroth-order optimization strategy to estimate the gradient for updating soft prompts.

**Black-Box Tuning** Black-box tuning has always been a popular topic that can be widely used in practice. As an extension of the zero-order optimizer, it shows very strong potential in large models. Turner et al. (2021) learn a superior efficiency on the Bayesian optimization. Based on this, several methods are proposed for the Bayesian prediction and high-dimensional search (Daulton et al., 2022a;b). Wang et al. (2018b); Liu et al. (2020) explore that its applicable dimensions do not increase linearly with sampling. Furthermore, it also helps to maintain stability in the attack-defense tasks (Chen et al., 2017; Tu et al., 2019). Actually, recent advances in large models also reveal its huge potential in fine-tuning tasks. Sun et al. (2022c); Yu et al. (2023) learn its efficiency in the large language models. Zheng et al. (2023); Sun et al. (2023b); Qi & Zhang (2023); Oh et al. (2023) indicate that the tuned prompts also could largely improve the generality. As a promising, black-box tuning brings considerable benefits to the application of large models on low-computing users. Sun et al. (2022a;b); Li et al. (2023b) all demonstrate it could achieve high generalization efficiency in large models in various fields of studies with lower computing. However, their approaches failed to consider data privacy. Currently, many high-quality data may not be published due to data privacy, which limits the application scenario for black-box tuning. Therefore, we utilize the FL paradigm to propose the Fed-BBPL which solves the issue of black-box service and data privacy.

# 3 FEDERATED PROMPT TUNING FOR BLACK-BOX MODEL

In this section, we mainly introduce the implementation paradigm of our proposed Fed-BBPL framework. We consider the general minimization problem with the global objective as:

$$\min_{\theta} F(\theta) = \frac{1}{m} \sum_{i=1}^{m} F_i(\theta), \ F_i(\theta) \triangleq \mathcal{L}(\phi; g(\theta; x_i) \oplus x_i), \tag{1}$$

where $\mathcal{L}(\cdot) : \mathbb{R}^d \to \mathbb{R}$ denotes the loss function aligned with the specific task, $\phi$ denotes the frozen parameter set of the black-box model via API services, $g(\theta)$ denotes the prompt/instruction generator with learnable parameters $\theta$, and $x_i$ is the private high-quality dataset stored in user-$i$. $\oplus$ denotes the coupling of how prompts/instructions work on the raw data in different tasks.

Our target is to optimize a global generator that can fully benefit from the different dataset $x_i$ to improve the generalization ability of PTMs. We introduce the paradigm in Algorithm 1. At the beginning of each round, each user downloads the $\theta_t$ from the central server as the initialization state of the generator. Then it calculates the prompts and generates the coupling data $g(\theta_{i,t}; x_i) \oplus x_i$ locally. Then each user sends them to the PTM via API services. After the forward calculation, each user can receive the prediction and loss from PTM. According to the different tasks, each user adopts the corresponding black-box optimizer to update the parameters $\theta_{i,t}$. Repeat this process until the central server issues the activation. Then local

---

**Algorithm 1:** Fed-BBPL Algorithm

**Input:** initial parameters $\theta_0$ and $\phi$, $T$
**Output:** global generator $\theta_T$

1 **for** $t = 0, 1, \cdots, T-1$ **do**
2     download $\theta_t$ from server as $\theta_{i,t}$ parallelly
3     **while** *not activating by server* **do**
4         generate the prompts $g(\theta_{i,t}; x_i)$
5         send $g(\theta_{i,t}; x_i) \oplus x_i$ to PTM via API
6         receive $\mathcal{L}(\theta_{i,t}; x_i)$ from API
7         $\theta_{i,t} = ZeroOpt(\mathcal{L}, g(\theta_{i,t}), x_i)$
8     **end**
9     upload $\theta_{i,t}$ to the central server
10 **end**
11 aggregate all received $\theta_{i,t}$ as $\theta_{t+1}$

---

users send the optimized $\theta_{i,t}$ to the central server for aggregation. Specifically, in this work, we test our proposed Fed-BBPL in 3 classical tasks in the CV and NLP communities, i.e., image classification (Sec. 3.1), text classification (Sec. 3.2), and finding the best instruction (Sec. 3.3). Due to their variant objectives, the loss function $\mathcal{L}(\cdot)$, the generator $g(\cdot)$, and the *ZeroOpt* method are also a little different as shown in Table 2. We introduce their implementation details in the following part.

## 3.1 CV - IMAGE CLASSIFICATION

In the image classification task, given a fixed PTM $\phi$, we assume the prediction of the target is $P_\theta(y; x)$ where $x$ denotes the input samples. Let $y$ denote the label of data samples $x$, we consider the objective function as:

$$\min_{\theta} \left\{ \frac{1}{m} \sum_{i=1}^{m} -\log P_{\theta|\phi}(y; g(\theta, x_i) \oplus x_i) \right\}. \tag{2}$$

Table 2: Comparision of different tasks in our experiments.

| Task | Image Classification | Text Classification | Instruction Optimization |
|---|---|---|---|
| # Dataset | 12 | 7 | 21 |
| Codebase | BlackVIP | BDPL | InstructZero |
| Optimization | SPSA | VR-PGE | BO |
| Zeroth-order | ✓ | ✓ | ✓ |
| Generator | Encoder & Decoder | Categorical Distribution | Gaussian Distribution & LLM |
| Tuned Params. | Decoder | Param. of Distribution | Mean & Var. of Distribution |
| Prompt | Continuous | Continuous $\rightarrow$ Discrete | Continuous $\rightarrow$ Discrete |

Following BlackVIP (Oh et al., 2023), the lightweight prompt generator consists of a fixed encoder extractor and a tuned decoder generator with the parameters $\theta = \{\theta_e, \theta_g\}$. The function $g(\theta)$ is built from an autoencoder-based model. We freeze encoder parameters set $\theta_e$ from a pre-trained model on the ImageNet (Deng et al., 2009) dataset and only tune the $\theta_g$ on the local dataset, which is widely adopted in previous works. With the global aggregation, the encoder only has a light impact on the outputs. After the encoder, we concatenate the output and feed them into a convolutional decoder to generate prompts. We add scaled prompts to data samples as outputs and use a coefficient to control the ratio of prompts in synthetic samples (Neekhara et al., 2022; Oh et al., 2023). The details of visual prompt tuning and BlackVIP are illustrated in Appendix A.1.

**ZeroOpt.** Due to the model privacy in practice, we follow the Simultaneous Perturbation Stochastic Approximation (SPSA) (Spall, 1992; 1997) to approximate the high-demensional gradient efficiently. Let $\alpha \in (0, 1)$ be the perturbation step, we evaluate the update direction as:

$$\mathbf{g}_{zo} = \frac{\mathcal{L}(\theta + \alpha q) - \mathcal{L}(\theta - \alpha q)}{2\alpha} \cdot q^{-1}, \tag{3}$$

where $q \in \mathbf{R}^d$ is a $d$-dimensional random perturbation vector sampled from non-zero distributions. With this estimation, we implement the quasi-gradient updating process on local users.

### 3.2 NLP - TEXT CLASSIFICATION

For TC tasks, discrete prompt tuning aims to learn $n$ discrete prompt tokens $g(\theta) = p = p_1 p_2 ... p_n$ where $p_i$ is a word from vocabulary list $V$ with total $N$ tokens. We apply the BDPL (Diao et al., 2022), black-box discrete prompt learning, as our base black-box method. BDPL randomly samples $p_i$ from independent categorical distribution $p_i \sim \text{Cat}(\theta_i)$, where $\theta_i \in \{\theta : ||\theta||_1 = 1, 0 \leq \theta \leq 1\}$. Here, $\theta$ is the parameter required to be optimized. Following Diao et al. (2022), we can formulate the objective as:

$$\min_\theta F(\theta) \triangleq \frac{1}{m} \sum_{i=1}^m \mathbb{E}_{p \sim \text{Cat}(\theta)} [\mathcal{L}(p)] = \frac{1}{m} \sum_{i=1}^m \int \mathcal{L}(p) P(p) dp. \tag{4}$$

where $m$ is the sample size. Due to the discrete prompts sampling, we cannot directly update its parameter set as a continuous variable. Same as BDPL (Diao et al., 2022), we also utilize the policy gradient algorithm to update $\theta$.

**ZeroOpt.** By the expectation estimation of the gradient, we could update the parameter set with the quasi-gradient from a policy gradient estimator (PGE) as:

$$\mathbf{g}_{zo} = \nabla_{\theta_i} \mathbb{E}_{p \sim \text{Cat}(\theta)} [\mathcal{L}(p)] = \mathbb{E}_{P(p)} [\mathcal{L}(p) \nabla_{\theta_i} \log(P(p_i))]. \tag{5}$$

Thus we adopt the expectation to explicitly solve the gradient. Let $\theta_{i,j}$ be the $j$-th element in each sampling prompt, and let $j_i$ be the $j$-th element in each sampling probability, we can solve the quasi-gradient as $\nabla_{\theta_{i,j}} \log P(p_i) = \nabla_{\theta_{i,j}} \log \theta_{i,j_i}$ where $\sum_{j_i} \theta_{i,j_i} = 1$. For the $j$-th element, its quasi-gradient can be calculated by $1/\theta_{i,j}$ if $j = j_i$, otherwise $-1/\theta_{i,j}$ if $j \neq j_i$. Therefore, by receiving the output of the PTMs, we can update the coefficient $\theta$ with the current loss $\mathcal{L}(p)$.

### 3.3 NLP - FIND BEST INSTRUCTION.

As an instruction follower, current Large Language Models are often used as a black-box service which makes finding a better instruction more difficult than which-box models. InstructZero (Chen

et al., 2023) is a SOTA solution for this task using black-box prompt tuning. The optimization objection is to maximize the performance score $\mathcal{L}(p, x; y)$, where $g(\theta) = p = p_1 p_2 ... p_n$ is the instruction, $x$ is the input query and $y$ is the ground truth corresponding to $x$. Due to the discrete tokens of instructions and the black-box service, we apply an open-source LLM $R(\cdot)$ to convert soft prompt $\theta' \in \mathbb{R}^{d'}$ into a human-readable instruction $p$, where $d'$ is the dimension of the hidden state of the open-source LLM. To reduce the high dimension, we utilize a random linear projection matrix $A \in \mathbb{R}^{d \times d'}$, where $d \ll d'$ and $A$ is sampled from Uniform distribution (Wang et al., 2016). During the training, $A$ is fixed without losing performance (Kleinberg, 1997; Chen et al., 2023).

To efficiently optimize the projection parameters, we utilize Bayesian optimization (BO) to update the soft prompts. The final objective could be formulated as:

$$\max_{\theta} \frac{1}{m} \sum_{i=1}^{m} \mathcal{L}(\theta; x_i) \triangleq \frac{1}{m} \sum_{i=1}^{m} \mathbb{E}_{x_i} \left[ \mathcal{L}(R(A_i \theta'; x_i); x_i) \right]. \tag{6}$$

**ZeroOpt.** Let Gaussian processes (GP) be the prior of performance function $\mathcal{L}(\theta; x)$ with the mean function of $\mu(\theta)$ and variance function $\sigma(\theta)$. Given $m$ soft prompts $\theta$ and their performance $\mathcal{L}(\theta)$, we can further estimate a new posterior by maximizing an expected improvement acquisition function, i.e., $\mathbb{E}_{\mathcal{L} \sim \mathcal{N}(\mu, \sigma)}[\max \{0, \mathcal{L}(\theta) - \max_{i \in [m]} \mathcal{L}(\theta_i)\}]$ on each local user. Thus, the Bayesian update rules indicate the best new soft prompt can be searched by the maximization of the acquisition function (EI) as $\theta_{m+1} = \arg \max \mu(\theta)$. The new best soft prompt $\theta_{m+1}$ will be utilized to update the mean and variance of the GP for the next training iteration. Because the final goal is to optimize human-readable instructions, instruction-coupled kernel (Chen et al., 2023) is adopted for reflecting the similarity of the instruction generated by open-source LLM in the task. Each local loss function is approximated by an independent Gaussian process. Therefore, for the FL paradigm, all users can aggregate the parameters, AKA the soft prompt $\theta$ for potential improvements from the other local dataset. More details are shown in Appendix C.

## 4 EXPERIMENTS

We conduct three main experiments including Image Classification (Sec. 4.1), Text Classification (Sec. 4.2), and Finding Best Instructions on LLM (Sec. C.3). For each, we delineate the experimental configurations before presenting the primary findings and subsequent ablation analyses.

### 4.1 IMAGE CLASSIFICATION

The CLIP ViT-B/16 (Radford et al., 2021) is engaged as a black-box model service. For the prompt generator, we employ the `vit-mae-base` checkpoint pretrained on ImageNet. The zero-shot (ZS) results of CLIP are established as our baseline. Within the federated learning framework, we select either 5 or 10 clients and utilize three strategies for data partitioning: uniform distribution (IID), Dirichlet distribution (Dir), and the pathological strategy (Path). For the Dir-split, we reference Hsu et al. (2019) to incorporate a coefficient regulating its heterogeneity. In the Path-split, we curate an appropriate number of active categories based on the total classes. Herein, "Dir-X" and "Path-Y" signify that X% of active classes are preserved and that Y classes are retained, respectively. On the central server, the conventional aggregation termed *FedAvg* was employed, setting aside more intricate federated learning algorithms for future research. Notably, different from BlackVIP (Oh et al., 2023), our approach leans towards full-shot rather than few-shot training. Despite potential overfitting concerns inherent to full-shot settings, the outcomes underscore the viability of our proposed framework Fed-BBPT. Comprehensive details of experimental settings are shown in Appendix A.

### 4.1.1 RESULTS & ABLATION STUDY

Table 4 reveals that with a client count of 10, the proposed Fed-BBPT framework with the Path-10 data splitting surpasses than the zero-shot baseline by 2.2%. Our approach proves superior or at par across 11 datasets, underlining the potency of federated black-box prompt tuning. Remarkably, of all data partitioning strategies, Path-10 outperforms IID and Dir-0.3 sampling. The varying dominance of different strategies across datasets reiterates the versatility of our Fed-BBPT framework. For intricate datasets such as Clevr which necessitates visual reasoning (Oh et al., 2023), Fed-BBPT

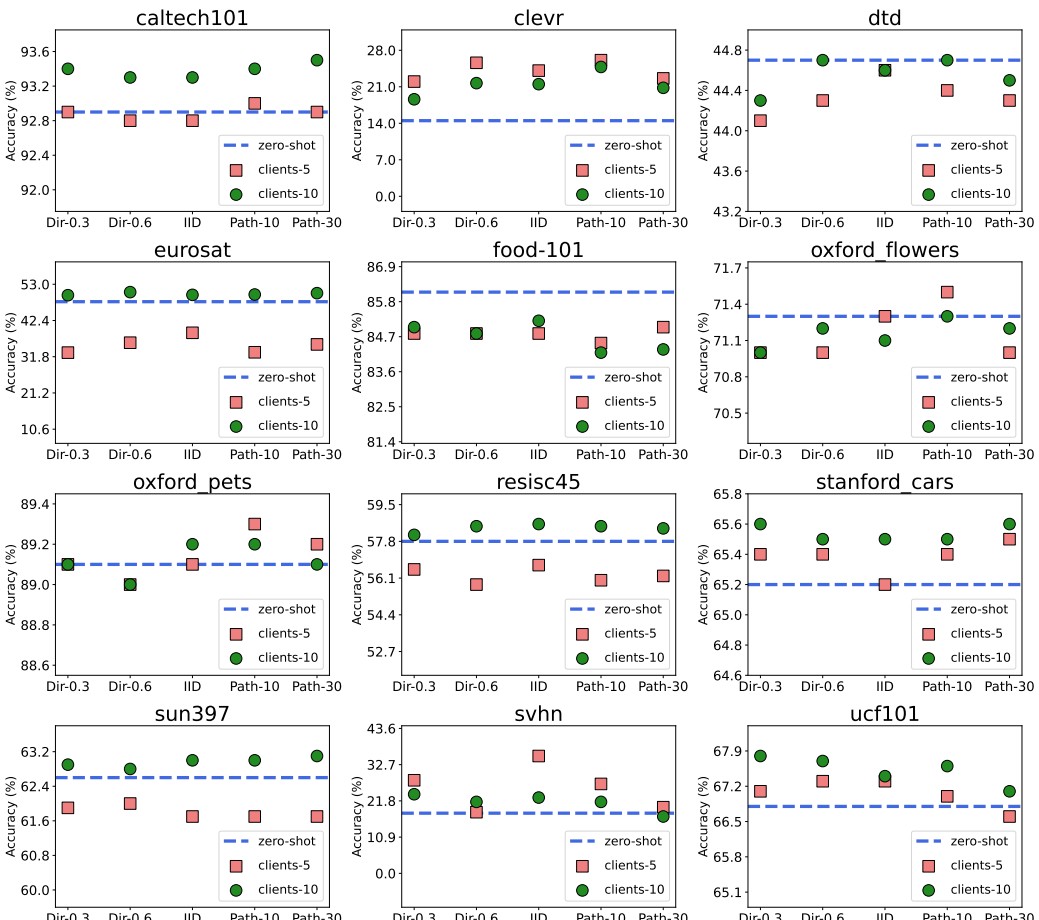

Figure 2: The effects of different hyperparameters for dividing datasets and number of clients across all 12 datasets. All experiments are conducted under full-shot settings.

grounded on BlackVIP still registers significant enhancements over the zero-shot baseline. The modest outcomes and the inability to markedly eclipse the baseline for the food dataset can likely be ascribed to overfitting inherent to the full-shot training paradigm.

Further experiments explored variations in the number of clients and the nuances of the division strategies. As inferred from Table 3, in the full-shot setting, ten clients, on average, outperformed five. Given the full-shot context, a larger client base implies lesser data per client, thereby potentially curtailing overfitting. As

Table 3: The average accuracy across 12 datasets on different number of clients 5 & 10.

| Clients | Dir-0.3 | Dir-0.6 | IID | Path-10 | Path-30 |
|---------|---------|---------|------|---------|---------|
| 5 | 59.8 | 59.7 | **61.0** | 59.9 | 59.2 |
| 10 | **60.8** | **61.0** | **61.0** | **61.1** | **60.4** |

shown in Fig. 2, the configuration with ten clients triumphed over its five-client counterpart in approximately eight datasets. In other datasets, the performances were largely comparable, save for Clevr. The anomalous behavior of Clevr can be attributed to its complexity, stemming from the need for visual reasoning capabilities (Oh et al., 2023). Furthermore, both ten-client and five-client configurations surpassed the zero-shot benchmark in a majority of the datasets, further accentuating the efficacy of our framework. The ablation studies also intimate that the method of data partitioning had a minimal bearing on outcomes, reconfirming the robustness of the Fed-BBPT framework.

## 4.2 TEXT CLASSIFICATION

For the realm of text classification, we adopt the Black-box Discrete Prompt Learning (BDPL) (Diao et al., 2022) as the fundational baseline and primary codebase. The black-box API model chosen for

Table 4: This table shows the classification accuracy across 12 datasets. The number of clients is 10. VP is the white-box baseline. ZS means zero-shot using CLIP which is the baseline. Approximately 10% of active classes are maintained in Dir-0.3 split and fixed 10 classes are maintained in Path-10 on each local user. All the results are the average of three different random seeds.

| Method | Caltech | Clevr | DTD | Eurosat | Food | Flower | Pet | Resisc | Cars | SUN | SVHN | UCF | *Avg.* |
|--------|---------|-------|------|---------|------|--------|------|--------|------|------|------|------|------|
| VP | 94.2 | 40.8 | 61.9 | 90.8 | 81.8 | 86.9 | 90.2 | 81.4 | 66.9 | 67.1 | 60.4 | 74.2 | 74.7 |
| ZS | 92.9 | 14.5 | **44.7** | 47.9 | **86.1** | **71.3** | 89.1 | 57.8 | 65.2 | 62.6 | 18.1 | 66.8 | 59.8 |
| IID | 93.3 | 21.5 | 44.6 | 49.9 | 85.2 | 71.1 | **89.2** | **58.6** | 65.5 | 63.0 | 22.8 | 67.4 | 61.0 |
| Dir-0.3 | **93.4** | 18.6 | 44.3 | 49.8 | 85 | 71.0 | 89.1 | 58.1 | 65.6 | 62.9 | **23.8** | **67.8** | 60.8 |
| Path-10 | **93.4** | **24.8** | **44.7** | **50.0** | 84.2 | **71.3** | **89.2** | 58.5 | **65.5** | **63.0** | 21.5 | 67.6 | **61.1** |

this experiment is RoBERTa-large (Liu et al., 2019). Given that the parameters of RoBERTa-large are accessible, we incorporate several white-box baselines for comparison, including finetuning (FT), Prompt Tuning (Lester et al., 2021), P-Tuning v2 (Liu et al., 2021), AutoPrompt (Shin et al., 2020) and FeatureProbe (Peters et al., 2019). On the black-box front, baselines include Manual-Prompt (Diao et al., 2022), InContextLearning (ICT) (Brown et al., 2020), BBT (Sun et al., 2022c), RLPrompt (Deng et al., 2022) and BDPL (Diao et al., 2022). Two sets of BDPL outcomes were juxtaposed; one sourced from the original BDPL publication and the other derived from our independent experiments. The experiments are conducted in the few-shot setting, constraining each client to minimal training examples to mitigate overfitting. A total of five local clients were incorporated, without compromising the general outlook. Analogous to the image classification experiments, we implement the *FedAvg* aggregation approach. Comprehensive details encompassing the datasets we use, metrics, and experimental hyperparameters are cataloged in Appendix B.

### 4.2.1 RESULTS & ABLATION STUDY

The overall outcomes across seven binary classification datasets are shown in Table 5. Primarily, our proposed approach, Fed-BBPT, conspicuously surpasses manually-crafted prompts by a margin of 10.1% and consistently outperforms black-box counterparts like ICT, BBT, and RLPrompt, with BDPL being the sole exception. Moreover, Fed-BBPT showcases superiority over a plethora of both white-box and black-box baselines except FT and FeatureProbe. Notably, given that all black-box white-box methods trail behind the FT outcomes and Fed-BBPT outdoes FeatureProbe in four out of seven datasets, the overall efficacy of Fed-BBPT remains evident. Furthermore, Fed-BBPT achieves the highest scores in datasets such as SST-2, QNLI, and RTE, and exhibited competitive performance in CoLA, MRPC, and QQP. In the context of the MNLI dataset, utilizing the BDPL codebase, our results did not align with those presented in the original BDPL study. However, Fed-BBPT's performance on MNLI still outstripped our standalone BDPL rendition. This consistent trend accentuates the proficiency of our proposed federated framework in text classification tasks.

For a deeper analytical perspective, we manipulate the client count and training shots to execute ablation studies on two datasets: SST-2 and MRPC. The findings are illustrate in Fig. 5 located in Appendix B.3. It shows that with consistent total training shots, Fed-BBPT persistently outperformed BDPL. In both datasets, the generalization efficacy exhibits a marked augmentation with the expansion of the client ensemble up to a specified threshold, beyond which overfitting phenomena manifest, culminating in a deterioration of performance. This robustly underscores the notion that amalgamated insights from an array of local users can further bolster the adaptability of PTMs.

### 4.3 INSTRUCTION TUNING ON LARGE LANGUAGE MODEL

Parameter-efficient instruction tuning has drawn much attention with the development of large language models (Zhang et al., 2023b). Considering the data privacy problem, applying federated learning to instruct tuning the LLMs is proposed by Zhang et al. (2023a). However, the previous approaches fail to consider the huge cost of storing the LLM on local users which limits the application of federated instruction tuning. Inspired by the InstructZero (Chen et al., 2023) which utilizes a black-box tuning strategy to generate better instruction locally, we apply our proposed Fed-BBPT framework on it, without storing the whole LLM while protecting the data privacy. Please see Appendix C for more details of the model architecture and experimental setting.

Table 5: This table shows the metric scores across 7 text datasets. The number of clients is 5. BDPL denotes the results reported in the original paper while BDPL* represents the results we get from running the published code by ourselves. For the federated approach, all clients are in 8-shot settings, while BDPL* is under 16-shot settings.

| Datasets | SST-2 | CoLA | MNLI | MRPC | QNLI | QQP | RTE | *avg.* |
|---|---|---|---|---|---|---|---|---|
| | | | *White-Box Methods* | | | | | |
| FT | $86.5_{\pm2.0}$ | $20.4_{\pm1.9}$ | $50.8_{\pm1.2}$ | $78.4_{\pm1.3}$ | $53.2_{\pm1.8}$ | $60.8_{\pm1.9}$ | $55.6_{\pm2.5}$ | 58.0 |
| PromptTuning | $70.7_{\pm2.6}$ | $8.0_{\pm0.7}$ | $36.5_{\pm0.9}$ | $52.7_{\pm3.4}$ | $53.5_{\pm1.6}$ | $50.2_{\pm1.5}$ | $56.3_{\pm1.6}$ | 46.8 |
| P-Tuning v2 | $80.4_{\pm1.2}$ | $8.9_{\pm2.7}$ | $44.2_{\pm1.7}$ | $62.4_{\pm2.0}$ | $51.5_{\pm1.3}$ | $57.4_{\pm2.4}$ | $53.1_{\pm1.7}$ | 51.1 |
| AutoPrompt | $71.5_{\pm2.1}$ | $5.4_{\pm2.3}$ | $40.1_{\pm1.5}$ | $63.8_{\pm3.1}$ | $50.2_{\pm1.3}$ | $45.7_{\pm1.3}$ | $52.1_{\pm1.6}$ | 47.0 |
| FeatureProbe | $79.5_{\pm1.6}$ | $15.6_{\pm1.2}$ | $46.5_{\pm1.8}$ | $68.9_{\pm1.7}$ | $50.5_{\pm0.2}$ | $56.3_{\pm1.1}$ | $54.1_{\pm2.5}$ | 53.1 |
| | | | *Black-Box Methods* | | | | | |
| ManualPrompt | $77.2_{\pm2.1}$ | $0.6_{\pm0.0}$ | $35.9_{\pm1.3}$ | $70.4_{\pm1.6}$ | $49.2_{\pm1.1}$ | $49.8_{\pm0.9}$ | $48.2_{\pm0.6}$ | 47.3 |
| ICT | $82.8_{\pm2.1}$ | $1.1_{\pm0.4}$ | $37.2_{\pm1.6}$ | $72.1_{\pm2.3}$ | $50.8_{\pm0.5}$ | $50.1_{\pm0.9}$ | $49.3_{\pm2.3}$ | 49.1 |
| BBT | $85.3_{\pm3.9}$ | $5.5_{\pm2.7}$ | $40.6_{\pm2.5}$ | $66.4_{\pm3.7}$ | $55.4_{\pm3.2}$ | $55.2_{\pm3.1}$ | $52.6_{\pm2.2}$ | 51.6 |
| RLPrompt | $88.4_{\pm1.9}$ | $5.0_{\pm1.1}$ | $42.8_{\pm3.2}$ | $68.9_{\pm2.1}$ | $52.6_{\pm1.4}$ | $53.7_{\pm2.2}$ | $51.8_{\pm1.8}$ | 51.8 |
| BDPL | $87.6_{\pm2.1}$ | $4.6_{\pm1.2}$ | $42.5_{\pm1.8}$ | $78.1_{\pm3.7}$ | $53.1_{\pm1.1}$ | $56.4_{\pm1.9}$ | $53.5_{\pm1.9}$ | 53.7 |
| BDPL* | $89.3_{\pm2.0}$ | $6.4_{\pm1.8}$ | $29.1_{\pm2.5}$ | $77.4_{\pm1.9}$ | $55.9_{\pm1.0}$ | $54.9_{\pm1.5}$ | $56.0_{\pm1.9}$ | 52.7 |
| | | | *Federated Black-Box Method* | | | | | |
| Fed-BBPT | $\mathbf{89.5}_{\pm0.5}$ | $4.6_{\pm0.6}$ | $29.4_{\pm1.3}$ | $75.1_{\pm4.0}$ | $\mathbf{56.3}_{\pm1.2}$ | $53.8_{\pm0.0}$ | $\mathbf{56.1}_{\pm2.3}$ | 52.1 |

### 4.3.1 RESULTS

The experimental outcomes for APE, Uniform, InstructZero, and Fed-BBPT across 21 datasets are depicted in Fig. C.3. Notably, Fed-BBPT consistently outperforms or matches the InstructZero method in the majority of the datasets, underscoring the potential of our proposed framework. For datasets where a perfect score of 1 was achieved, such as "Letters List" and "Sum", Fed-BBPT matched this score. In tasks where InstructZero underperformed, including "Translation EN-DE", "Categorization", and "Negation", Fed-BBPT exhibited superior results. Remarkably, for "Rhymes", "Word Sorting" and "Second Letter", the performance gains were substantial, with improvements of 91%, 45.3%, and 29% respectively. These observations emphasize that federated learning can significantly enhance performance on challenging tasks. While Fed-BBPT mitigated performance decline in datasets like "Antonyms" and "Synonyms", the average results affirm that our Fed-BBPT framework is not only competitive with existing benchmarks but also addresses data privacy concerns.

## 5 CONCLUSION

In this study, we address three paramount challenges faced by practical applications of current large PTMs, especially for users with low computational resources: memory constraints, model privacy, and data privacy. To this end, we introduce an innovative framework, named Fed-BBPT, to facilitate federated black-box prompt tuning. The central feature of Fed-BBPT is its capability to harness consensus knowledge from multiple high-quality private datasets. It achieves this by periodically aggregating local learnable parameters, significantly enhancing the generalization ability of PTMs for various downstream tasks. A salient characteristic of Fed-BBPT is its optimizer-agnostic nature. It demonstrates broad applicability to several state-of-the-art (SOTA) black-box methods. To validate the effectiveness of our framework, we have carried out comprehensive experiments across a range of classical tasks in both CV and NLP. Fed-BBPT often yields better or comparable performance comparable to strong white-box and black-box baselines.

However, this study is not without its limitations. Firstly, our experiments primarily focus on rudimentary CV and NLP tasks. Secondly, we confine our implementation to a basic federated learning strategy, specifically FedAVG. As a future direction, we aspire to evaluate Fed-BBPT on more intricate datasets and benchmark it against a variety of baselines and methodologies. Additionally, we are keen on devising and incorporating advanced federated learning strategies to further elevate performance and ensure greater stability.

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

# A  IMAGE CLASSIFICATION

## A.1  FEDERATED BLACKVIP FRAMEWORK

In this section, we briefly introduce the BlackVIP and how to apply federated learning. The architecture is shown in Fig. 3 Bahng et al. (2022) first introduced the pixel-level *visual prompting* (VP) for pre-trained vision models. With a frozen PTM, VP, attached to input images, are tuned during training to adapt to a target dataset. During inference, the same VP concatenate all input images.

We utilize an encoder-decoder architecture as the prompt generator. Denote $p$ as the visual prompt generated by prompt generator $g(\theta_g)$ parameterized by $\theta_g = \{\theta_d, \theta_t\} \in \mathbb{R}^d$, where $\theta_d$ is decoder's parameter and $\theta_t$ is a task specific *prompt trigger vector* which is jointly optimized. Specifically, the input image $\widetilde{x}$ is:

$$\widetilde{x} = \text{clip}(x + \epsilon g(\theta_g, z_x)) \tag{7}$$

where $z_x = f(x)$ is the feature vector of input $x$ generated by the frozen encoder $f(\cdot)$, and $\epsilon \in [0, 1]$ is a hyperparameter that adjusts the influence of the visual prompt. CLIP is the black-box model. We apply SPSA as the optimization method as depicted in Sec. 3.1. In Fed-BBPT framework, we average the updated $\theta_d$ and $\theta_t$ after each epoch of training in central server, and then forward to each local server for the continual training steps. We also keep the SPSA with Gradient Correction (SPSA-GC) as our optimization method, whose details are discussed in Oh et al. (2023) Sec. 4.2.

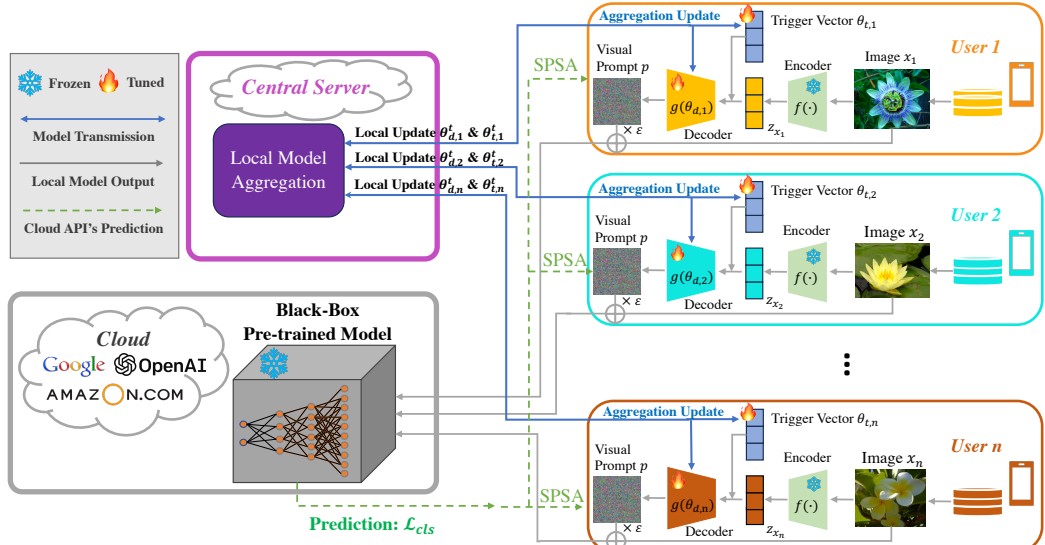

Figure 3: An overview of federated BlackVIP framework. Note that in tunable trigger vector $\theta_{t,i}^t$, the upper $t$ represents the iteration step while the lower $t$ means the "trigger vector", and the $i$ represents the client number. SPSA is the optimization method we use in this task.

## A.2  EXPERIMENT SETTINGS

We conduct the experiments on 12 widely used datasets: Caltech101 (Fei-Fei et al., 2004), Oxford-Pets (Parkhi et al., 2012), StanfordCars (Krause et al., 2013), Flowers102 (Nilsback & Zisserman, 2008), Food101 (Bossard et al., 2014), SUN397 (Xiao et al., 2010), DTD (Cimpoi et al., 2014), SVHN (Netzer et al., 2011) , EuroSAT (Helber et al., 2019), Resisc45 (Cheng et al., 2017), CLEVR (Johnson et al., 2017), UCF101 (Soomro et al., 2012). Those datasets cover different domains which require different ability of models such as visual understanding of actions or textures.

Following BlackVIP (Oh et al., 2023), we set the batch size as 128 for all datasets. The output dimension of the prompt generator's encoder is $N \times 768$, where $N$ is the batch size. We use *depth-wise separable convolution* (DSC) layer (Chollet, 2017) as the lightweight prompt generator's decoder. We set 800 as the dimension of prompt trigger vector, which is concated by the output of

encoder. The concated 1568-dimension vector will be shaped to a $32 \times 7 \times 7$ shaped 3D tensor as the input of decoder. Without any change from the original codebase, we apply SDG as the optimizer with learning rate 0.5 and momentum 0.9. We set $\epsilon$ as 1 and all other hyperparameters about SPSA same as the default parameter in the codebase of BlackVIP (Oh et al., 2023).

# B  TEXT CLASSIFICATION

## B.1  FEDERATED BDPL FRAMEWORK

BDPL Diao et al. (2022) apply Categorical distributions parameterized by $\theta$ as the discrete prompt generator. Therefore, we average the $\theta_i$ from all local server after each training epoch, then the averaged parameter was forwarded back to each local server as the initial state for the next training step. The framework is shown in Fig. 4.

BDPL consturcts the prompt candidates $V$ with $N$ tokens utilizing pointwise mutual information (PMI), inspired by Diao et al. (2021). PMI score evaluates the possibility of two words occurs at the same time. The higher the score is, the more likely the two words to form an n-gram. Specifically, PMI is calculated by

$$\text{PMI}(\overline{x}, \widetilde{x}) = \log \frac{p(\overline{x}\widetilde{x})}{p(\overline{x})p(\widetilde{x})} \tag{8}$$

where $\overline{x}$ and $\widetilde{x}$ are two adjacent words in the sentence, and $p(x)$ represents the probability of an n-gram $x$ (Diao et al., 2022). A delimiter will be inserted between the two words $\overline{x}$ and $\widetilde{x}$ when the PMI score is below a threshold $\sigma$. Finally, we form a list of ngrams $V$ after segment the sentences in the dataset.

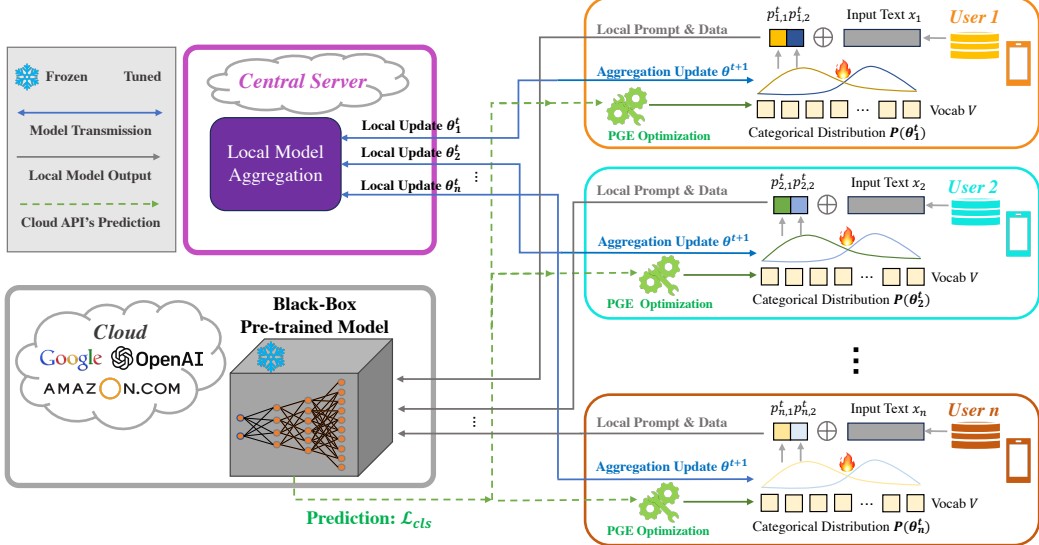

Figure 4: An overview of federated BDPL framework. The discrete prompt $p_{i,j}^t$ is sampled from prompt token vocabulary $V$, where $t$ is the training epoch, $i$ and $j$ represents the $i$-th client and the $j$-th prompt token.

## B.2  EXPERIMENT SETTINGS

For text classification task, we conduct experiments on seven binary classification datasets from GLUE benchmark (Wang et al., 2018a), including SST-2 (Socher et al., 2013), CoLA (Warstadt et al., 2019), MNLI (Williams et al., 2017), MRPC (Dolan & Brockett, 2005), QNLI (Wang et al., 2018a), RTE (Dagan et al., 2005; Haim et al., 2006; Giampiccolo et al., 2007; Bentivogli et al., 2009). Table 6 indicates the details of each datasets including the number of train/val/test samples, matrics, and the template we adpot.

We adopt AdamW to optimize the prompt generator for 30 epochs with the weight decay 0.1. The best learning rate, sample size and prompt length are different across datasets. We search the learning rate from $\{1e-4, 1e-5, 1e-6\}$, sample size from $\{10, 20, 30\}$ and prompt length form $\{10, 30, 50, 100\}$. The specific setting are shown in Table B.3. The batch size for training and evaluation is 128 and 64 respectively. we keep $N$ as 200 and set the API call limitation as 8000. We leave all other hyperparameters same as the original BDPL codebase (Diao et al., 2022).

Table 6: The statistics of seven datasets from GLUE benchmark we use in our experiments. C. is the number of classes for each classification dataset. MC. in metric represents the Matthews Correlation. The template depitcs the prompts and labels used in ManualPrompt method (Gao et al., 2020). Note that we apply few-shot setting as described in Sec. 4.2.

| Dataset | C. | Train | Val | Test | Metric | Template |
|---|---|---|---|---|---|---|
| MNLI | 3 | 393K | 9.8K | 9.8K | acc. | $sentence_1$ entailment? [MASK], $sentence_2$. (yes/no) |
| QNLI | 2 | 105K | 5.5K | 5.5K | acc. | $sentence_1$ entailment? [MASK], $sentence_2$. (yes/no) |
| RTE | 2 | 2.5K | 277 | 3K | acc. | $sentence_1$ entailment? [MASK], $sentence_2$. (yes/no) |
| SST-2 | 2 | 6.7K | 872 | 1.8K | acc. | $sentence_1$. It was [MASK]. (great/terrible) |
| MRPC | 2 | 3.7K | 408 | 1.7K | F1 | $sentence_1$. ?[MASK], $sentence_2$. (yes/no) |
| QQP | 2 | 364K | 40K | 391K | F1 | $sentence_1$ ?[MASK] $sentence_2$. (yes/no) |
| CoLA | 2 | 8.6K | 1K | 1K | MC. | $sentence_1$. correct? [MASK]. (yes/no) |

Table 7: The hyperparameters we choose for each dataset.

| Dataset | SST-2 | CoLA | MNLI | MRPC | QNLI | QQP | RTE |
|---|---|---|---|---|---|---|---|
| Sample Size | 10 | 10 | 30 | 10 | 20 | 30 | 30 |
| Learning Rate | $1e-4$ | $1e-4$ | $1e-4$ | $1e-4$ | $1e-5$ | $1e-5$ | $1e-5$ |
| Prompt Length | 100 | 50 | 10 | 30 | 50 | 10 | 10 |

## B.3 RESULTS AND ABLATION STUDY

We conduct ablation study on SST-2 and MRPC with different number of clients and the number of total training shots. For five clients, if the total number is ten, each client will be arranged by two shots randomly. We set ManualPrompt as the baseline. The details are shown in Fig. 5 with the discussion described in Sec. 4.2.1.

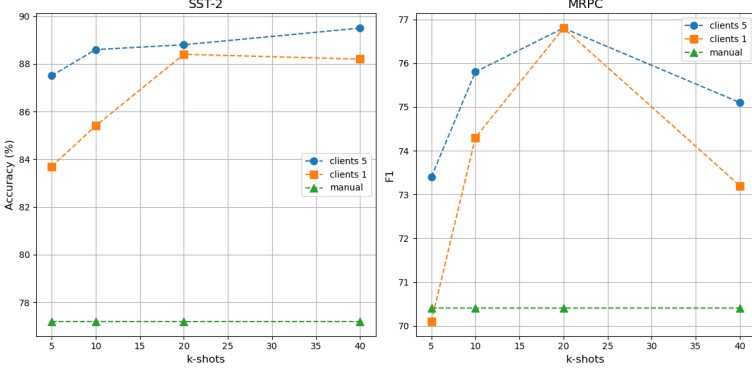

Figure 5: The results of ablation study on SST-2 and MRPC.

## C FIND BEST INSTRUCTION

### C.1 FEDERATED INSTRUCTZERO FRAMEWORK

This section shows the overall federated InstructZero framework in Fig. 6 and describes the details of bayesian optimization of soft prompt with instruction-coupled kernel (Honovich et al., 2022). Given the tunable soft prompt $\theta \in \mathbb{R}^d$ and the black-box objective function $\mathcal{L}(\theta)$, the goal is to update a posterior of $\mathcal{L}(\cdot)$ based on $(\theta, \mathcal{L}(\theta))$ by finding new best soft prompt $p$ untile the $\mathcal{L}$ converges.

We employ Bayesian Optimization (BO) for the updating process. Gaussian Process is applied as the prior distribution for the black-box objective value function $\mathcal{L}(\cdot)$, which can be parameterized by a mean function $\mu(\cdot) = 0$ and a covariance (kernel) function $k(\cdot, \cdot)$. Denote the soft prompts collected from all previous BO steps and their evaluation as $\boldsymbol{\theta}_{1:m} \triangleq \{\theta^1, \cdots \theta^m\}$ and $\mathcal{L}_{1:m} \triangleq \{\mathcal{L}(\theta^1), \cdots, \mathcal{L}(\theta^m)\}$ respectively. Then the mean and variance of a (GP) can be formulated by

$$\mu(\boldsymbol{\theta}) \triangleq \boldsymbol{k} \left( \boldsymbol{K} + \eta^2 \boldsymbol{I} \right)^{-1} \mathcal{L}_{1:m} \tag{9}$$

$$\sigma^2(\boldsymbol{\theta}) \triangleq k(\boldsymbol{\theta}, \boldsymbol{\theta}) - \boldsymbol{k}^\top \left( \boldsymbol{K} + \eta^2 \boldsymbol{I} \right)^{-1} \boldsymbol{k} \tag{10}$$

where $\boldsymbol{k} = [k(\boldsymbol{\theta}, \theta^1), \cdots, k(\boldsymbol{\theta}, \theta^m)]$ and $\eta$ represents the measurement of observations' noise levels. As described in Sec. 3.3, BO select the best soft prompt $\theta^{m+1}$ using expected improvement acquisition function (EI):

$$\theta^{m+1} \in \underset{\theta \in \mathbb{R}^d}{\arg\max}\, u(\theta) = \mathbb{E}_{\mathcal{L} \sim \mathcal{N}(\mu, \sigma)} \max \left[ \left\{ 0, \mathcal{L}(\theta) - \max_{i \in [m]} \mathcal{L}(\theta_i) \right\} \right] \tag{11}$$

The open-source LLM take the new best soft prompt $\theta^{m+1}$ concatenating with the task-specific in-context instruction $(\widetilde{x}_i, \widetilde{y}_i)_{i=1}^t$ as inputs and generate the new instruction $p^{m+1}$, which is evaluated by the black-box LLM on the target task. The evaluation score is $\mathcal{L}(\theta^{m+1})$. Then, BO replace the training data $(\boldsymbol{\theta}_{1:m}, \mathcal{L}_{1:m})$ with $(\theta^{m+1}, \mathcal{L}(\theta^{m+1})$ and repeat updating in Eq. (9)-(11). Chen et al. (2023) also introduces the Instruction-Coupled kernel which take the instruction similarity into consideration. For more details about this part, please refer to its original paper.

In the federated learning setting, every time when we get the best new soft prompt, we send the prompt and its evaluation score from local to central server. After averaging both, the prompt and evaluation score are forwarded back to local user for the next training step.

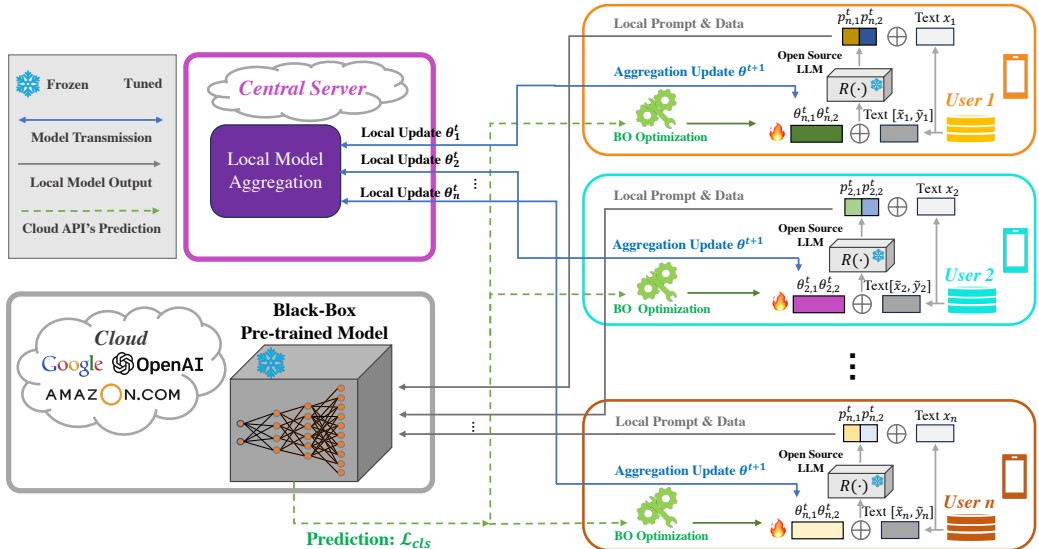

Figure 6: An overview of federated InstructZero framework. $R(\cdot)$ represents an open-source LLM which takes soft prompt (i.e. tuned parameters) as input and generate human-readable instructions $\boldsymbol{p} = p_1 p_2 ... p_n$. In the graph, the notion $i$ in $x_i$ and $y_i$ represents the number of client.

## C.2 EXPERIMENT SETTINGS

Using the InstructZero codebase, we conduct our experiments on 21 datasets randomly chosen from Honovich et al. (2022) and Chen et al. (2023). APE (Zhou et al., 2022b), uniform sampling, and InstructZero (Chen et al., 2023) are three baselines. WizardLM (Xu et al., 2023) is employed as the LLM prompt generator. Same as all previous experiments, we set five clients in total. For each client, we allocate 5 and 20 samples to the training and validation sets, respectively. In instances where the total data samples are insufficient for all clients, we distribute the maximum available samples equitably among them. The checkpoint we use for the open-source LLM is WizardLM-13B-V1.1 (Xu et al., 2023). We set the batch size as 10 which means that at every iteration, 10 soft prompts are explored. The dimensionality $d$ of soft prompt $\theta$ is set to 10.

## C.3 RESULTS

We report our experimental results in Fig. C.3.

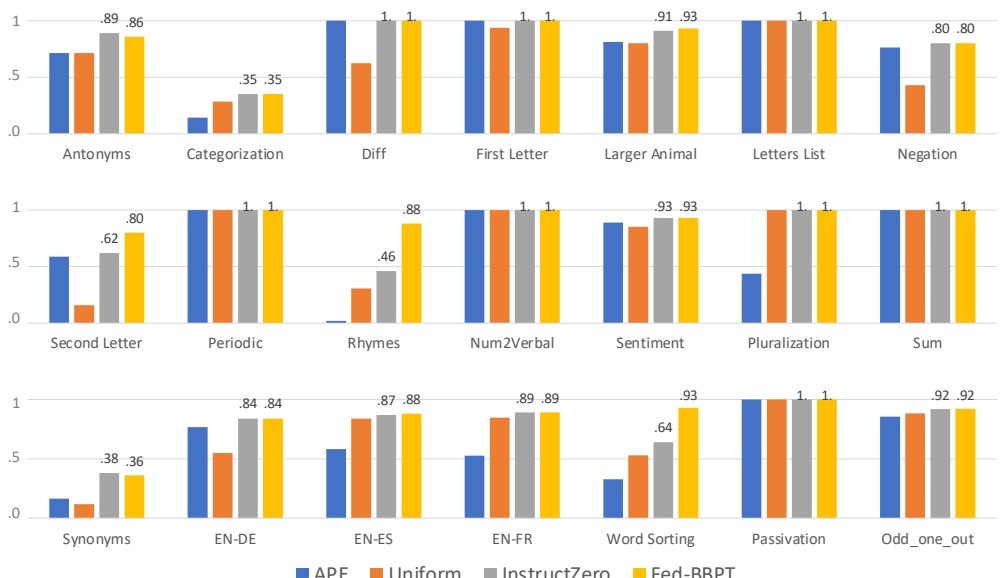

Figure 7: The results of 21 datasets randomly chosen from the InstructZero (Chen et al., 2023).

