# OpenReview forum: "Federated Tuning for Black Box Large Models"
_ICLR.cc/2024/Conference — ICLR 2024 Conference Withdrawn Submission_

### Official Review · Reviewer_viQp · 2023-10-28

**Soundness:** 2 fair
**Presentation:** 2 fair
**Contribution:** 2 fair
**Rating:** 3
**Confidence:** 4

**Summary:**

This paper considers a new setting: federated prompt tuning on black-box models to mitigate the issues of memory constraint and model privacy. Federated Black-Box Prompt Tuning (Fed-BBPT) is proposed, which covers three specific methods for three tasks, respectively. Fed-BBPT collaboratively trains the prompt generator under the coodination of a cloud server. At the same time, another cloud model provider receives the uploaded content from clients as the inputs for the pre-trained model, where this content may be composed by the generated prompt concatenate with raw text, and provides feedback to clients.

**Strengths:**

- This paper considers a new setting: federated prompt fine-tuning on black-box models.
- This paper considers three tasks and proposes one method for each task.
- This paper conducts experiments on multiple datasets.

**Weaknesses:**

- **Violation of data privacy.** Though this paper claims to protect data privacy, this claim is open to question and there needs some discussion on privacy. When it comes to preserving data privacy, one basic requirement is that the data should be kept locally (as what conventional FedAvg does). However, this method uploads the private data (in various formats for different tasks) to the cloud (the holder of pre-trained model). Isn't this a breach of privacy? In Figure 3, it is likely that an attacker can recover the raw image since the attacker can access to all the model parameters (decoder, encoder, ...). What's worse, in Figure 4 and 5, the **raw texts of clients are directly transmitted to the cloud.**
- The motivation should be further clarified. Nowadays, there are more and more open-source models such as CLIP and LLAMA. Is there a strong motivation to use FL on black-box models rather than FL on white-box (open-source) models given that the performance of FL on black-box models is quite limited?
- **Limited performance improvement**. The improvement brought by this method is quite limited given that the method has somewhat complex designs and still requires some resources. Take image classification task as an example. In Table 4, this method only outperforms zero-shot results by 1.0 ~ 1.3 on average  and cannot ensure consistent improvement (e.g., Food). However, from existing works, we know that few-shot fine-tuning can significantly improves the performance of CLIP [1] (the performance on Flowers102 can be improved to over 90% by 8-shot fine-tuning). Thus, there is a large room for improvement, which reflects that the improvement of this paper is quite minor. Besides, I suggest showing the performance of federated prompt learning over CLIP or FedCLIP [2], serving as a reference.
- The content in Section 3.2 is really confusing, especially the notations. $p_i$ is a word, $g(\theta)=p=p_1p_2...p_n$ indicates $p$ is a sequence of words, then what is the $p$ in Equation (4)? $p_i \sim Cat (\theta_i)$, what is $Cat$? $\theta_i \in \\{ \theta : ||\theta||_1 = 1, 0\leq \theta \leq 1\\}$, what is exactly $\theta$? In $g(\theta)$, $\theta$ seems to be a vector, but here what do you mean by $0\leq \theta \leq 1$? Also, what are exactly $\theta$, $\theta_i$ and $\theta_i,j$? There are so many confusions that I cannot understand this whole part.
- Missing experimental details. It is unclear how the baselines in Table 5 are implemented. What datasets do the baselines use? The union of clients' datasets or dataset of one client?

[1] Zhou, Kaiyang, et al. "Learning to prompt for vision-language models." International Journal of Computer Vision 130.9 (2022): 2337-2348.

[2] Wang Lu, Xixu Hu, Jindong Wang, and Xing Xie. Fedclip: Fast generalization and personalization for clip in federated learning. arXiv preprint arXiv:2302.13485, 2023.

**Questions:**

no

---

### Official Review · Reviewer_sZJ9 · 2023-10-31

**Soundness:** 3 good
**Presentation:** 3 good
**Contribution:** 2 fair
**Rating:** 5
**Confidence:** 5

**Summary:**

This paper extended black-box prompt tuning to federated learning scenarios, where they claimed the proposed method Fed-BBPT can address three challenges including model and data privacy, and memory constraint. They conducted extensive experiments to justify the effectiveness of their method.

**Strengths:**

This paper is clearly organized and presented. It is good to see a comparison in Table 1. The idea is direct and the algorithm is easy to understand. The experiments on different type of tasks are implemented with different forms of zero-order optimizations and results look good as well.

**Weaknesses:**

My major concerns are on data privacy (also related to technical contribution), efficiency, and research scope of this work.

**Questions:**

1.	The black-box tuning with data privacy concern is already mentioned in the existing work [1], and thus I felt that the claim at the end of Black-Box Tuning in page 4 is not precise.

        [1] Earning Extra Performance from Restrictive Feedbacks, 2023

2.	Table 1 showed that BlackVIP, BDPL, and InstrictZero are not preserving data privacy, because they are not under a FL framework. So, the data privacy is implemented via FL? In this sense, what kind of privacy is interested should be clarified. Otherwise, for example, what if I concerned that the pre-trained model is not trusted?

3.	The computing efficiency in Table 1 is based on freezing some model parameters and update prompts generator only. However, gradient estimation to $\theta$ is not efficient due to the cost of ZeroOpt. When compared with white-box models, it is necessary to state the additional computation burden. For example, how slow the proposed method is by comparing with VP in Table 4.

4.	It seems that the proposed method focuses on discriminative models only. However, the LLMs like OpenAI showed in Fig. 1 are often used as generative tasks. I was wondering if the proposed method is extendable to generative tasks.

5.	Following 2, I would concern the sufficiency of technical contributions of this work. Please clarify it if you do not think so.

---

### Official Review · Reviewer_NEfj · 2023-11-01

**Soundness:** 2 fair
**Presentation:** 2 fair
**Contribution:** 3 good
**Rating:** 6
**Confidence:** 3

**Summary:**

This paper points out three challenges when we finetune the pre-trained models in federated learning: (i) memory constraint, (ii) model privacy, and (iii) data privacy. To tackle these challenges, the authors propose Fed-BBPT, a black-box prompt tuning approach. Specifically, the clients optimize the prompt generator via a zero-order optimizer, which could differ among different tasks, and the server aggregates and updates the global generator. Extensive experiments cover CV and NLP tasks and validate the effectiveness of Fed-BBPT.

**Strengths:**

1. This paper explores how to fine-tune pretrained models in federated learning, which is a very interesting topic.
2. This paper covers three typical tasks using large pretrained models and discusses how to design the zeroth-order optimizer for each case.
3. The experimental results are intriguing when compared to other baselines.

**Weaknesses:**

1. The data privacy issue is not fully addressed because the training data needs to be submitted to the cloud via API.
2. As the training collaborates with the LLM providers, the monetary costs would be a big issue. Apparently, the more the clients call the API, the higher the costs they should afford.

**Questions:**

Please address my concerns listed in the weaknesses part.

---

### Official Review · Reviewer_aER3 · 2023-11-04

**Soundness:** 2 fair
**Presentation:** 3 good
**Contribution:** 2 fair
**Rating:** 3
**Confidence:** 4

**Summary:**

In this submission, the authors point out several challenges when applying pre-trained models (PTMs) in real-world applications, including memory constraint, model privacy, and data privacy, which motivates the authors to propose Federated Black-Box Prompt Tuning (Fed-BBPT). The proposed method adopts black-box tuning techniques in the context of federated learning. The authors provide three instantiations of Fed-BBPT in image classification, text classification and instruction optimization, and conduct experiments to compare the proposed method with several baselines.

**Strengths:**

1. This paper studies an interesting and practical problem, and provides meaningful discussions on applying PTMs.
2. The paper is well-written and easy to follow.
3. Three different instantiations of the proposed Fed-BBPT are provided in the paper, together with detailed implementations, which are really helpful for the community to reproduce.

**Weaknesses:**

1. The novelty of this paper is limited. As claimed by the authors, black-box tuning and zero-order optimization techniques are popular and have been widely used in real-world PTMs applications. This paper seems to provide a straightforward solution that directly applies these techniques in the context of federated learning. The authors can highlight the challenges of applying the existing techniques in new scenarios (e.g., FL) and how their design addresses the challenges.
2. The experiments are a little not convicting. For example, for the text classification tasks, the authors train RoBERTa-large on seven datasets from GLUE separately (please correct me if I have misunderstood something), with some naïve prompts provided in Table 6. These settings are far away from how we use LLMs nowadays (the model is finetuned one time and can be used in multiple tasks with different prompts) and more similar to a pretrain-finetune paradigm like BERT.
3. More depth analysis and discussions should be provided. For example, how to define and evaluate the "computing efficiency" and "memory efficiency" of Fed-BBPT compared to baselines (Table 1).

**Questions:**

Please refer to the weaknesses above.